

# Transcription factor 7-like 2 single nucleotide polymorphisms rs290487 and rs290481 are associated with dyslipidemia in the Balinese population

Prisca C. Limardi[1,2], Sukma Oktavianthi[1], Lidwina Priliani[1], Retno Lestari[2], Made Ratna Saraswati[3], Ketut Suastika[3] and Safarina G. Malik[1]

[1] Genome Diversity and Diseases Laboratory, Eijkman Institute for Molecular Biology, National Research and Innovation Agency, Jakarta, DKI Jakarta, Indonesia
[2] Department of Biology, Faculty of Mathematics and Natural Sciences, Universitas Indonesia, Depok, West Java, Indonesia
[3] Division of Endocrinology and Metabolism, Department of Internal Medicine, Faculty of Medicine, Udayana University, Denpasar, Bali, Indonesia

## ABSTRACT

**Background:** Dyslipidemia is one of the major risks for the development of cardiovascular diseases which has been the leading cause of death in developing countries. Previously, common polymorphisms of the transcription factor 7-like 2 (*TCF7L2*) gene have been associated with altered lipid profiles. In this study, we investigated the associations of *TCF7L2* SNPs, rs290487 and rs290481, with dyslipidemia and altered lipid profile in the Balinese.

**Methods:** A total of 565 subjects from four locations in the Bali Province, Indonesia, were recruited. Serum lipid concentrations (triglycerides (TG), low-density lipoprotein cholesterol (LDL-C), high-density lipoprotein cholesterol (HDL-C), total cholesterol (TC)) were measured using standard protocol. SNP genotyping was done using the amplification refractory system mutation polymerase chain reaction (ARMS-PCR) method.

**Results:** We found the shifted major/minor allele frequencies of both SNPs (0.56 for rs290487 T allele, 0.53 for rs290481 T allele) in the Balinese, as compared to dbSNP. The rs290487 and rs290481 C alleles were significantly associated with dyslipidemia, particularly high TC and high LDL-C. These associations were independent of age, sex, population, obesity, diabetes mellitus, and high TyG index as a proxy for insulin resistance. The haplotype CC also showed similar association with these traits. Our findings indicate that *TCF7L2* polymorphisms are associated with dyslipidemia and altered lipid profile in the Balinese.

# INTRODUCTION

Dyslipidemia is one of the risk factors for the development of cardiovascular diseases, the major cause of death in developing countries. Dyslipidemia refers to a condition of

Corresponding author
Safarina G. Malik,
safarina.malik@mrinstitute.org

abnormal lipid levels, including high triglycerides (TG), high low-density lipoprotein-cholesterol (LDL-C), low high-density lipoprotein-cholesterol (HDL-C), and high total cholesterol (TC) (*Lin et al., 2018*). According to the 2018 National Basic Health Survey, the national prevalence of high TG, high LDL-C, low HDL-C, high TC in Indonesia were 13.3%, 24.9%, 24.3%, 21.2%, respectively (*Kementerian Kesehatan, 2019a*). As a multifactorial disorder, dyslipidemia can occur due to interactions between genetic and environmental factors, such as dietary intake and lifestyle (*Cole, Nikpay & McPherson, 2015*).

One of the potential genetic risk factors for dyslipidemia is transcription factor 7-like 2 (*TCF7L2*) gene polymorphisms. *TCF7L2* encodes for a transcription factor containing the HMG-Box domain which plays a crucial role as the main effector of canonical wingless-type (Wnt) signaling pathway. The Wnt signaling pathway mostly regulates the expression of a wide array of metabolic genes, such as *PDK1* (*Pate et al., 2014*), *LGR4/5/6* (*Chen & Wang, 2018*), and *Gcg* (*Yi, Brubaker & Jin, 2005*). The *TCF7L2* pre-mRNA has 17 exons, including five alternative exons (exon 4, 13, 14, 15, 16) that will undergo alternative splicing mechanism, resulting in various mature tissue-specific mRNA isoforms (*Hansson et al., 2010*). *TCF7L2* has been widely studied for its association with metabolic-related diseases, such as type 2 diabetes mellitus (*Mayans et al., 2007*; *Villareal et al., 2010*; *Shokouhi et al., 2014*; *Zhu et al., 2017*; *Zhou et al., 2019*), obesity (*Al-Daghri et al., 2014*), metabolic syndrome (*DeMenna et al., 2014*), and cancers (*Folsom et al., 2008*; *Chen et al., 2013*) in various populations.

Due to westernization and urbanization in the past three decades, the Balinese population has undergone lifestyle changes that raised the prevalence of metabolic syndrome (MetS) and metabolic disorders, such as obesity and diabetes mellitus (DM). These traits are known to be closely associated with dyslipidemia through insulin resistance (*Athyros et al., 2011*; *Suastika et al., 2011*, *2019*), which also increase the Balinese's risk for dyslipidemia. A cross-sectional survey study carried out in Bali Island had reported the frequency of dyslipidemia in normal glucose tolerance subject group as follows, high LDL-C (73.8%), high non-HDL-C (53.9%), low HDL-C (31.3%), and high TG (20.4%). Meanwhile, both impaired fasting glycemia (IFG) and DM subject groups had relative higher percentage in all parameters, showing a positive correlation between glucose impairment with dyslipidemia (*Suastika et al., 2019*).

A prior study had found associations of *TCF7L2* common polymorphisms (rs7903146, rs1225372, and rs10885406) with altered lipid profiles in the Balinese. These findings indicated a relationship between *TCF7L2* and dyslipidemia. However, the minor allele frequencies of these three SNPs were rather low in Balinese, and thus they were hard to detect (*Oktavianthi et al., 2018*). Therefore, in the present study, we investigated the associations of other *TCF7L2* intronic SNPs, rs290487 (C > T, intron 8) (*NCBI, 2021a*) and rs290481 (C > T, intron 16) (*NCBI, 2021b*), which have higher allele frequencies in East-Asian (MAF > 0.60). Previously, rs290487 and rs290481 were reported to be associated with type 2 diabetes mellitus in the East-Asian population (*Chang et al., 2010*; *Luo et al., 2009*; *Zhu et al., 2017*), but their associations with dyslipidemia have not been well-studied. We hypothesize that *TCF7L2* SNPs rs290487 and rs290481 might also be

associated with dyslipidemia in Balinese population and could be used to predict the risk of dyslipidemia.

## MATERIALS AND METHODS

### Subjects, study design, and measurements

A cross-sectional study enrolling 565 unrelated subjects from four locations (Nusa Ceningan, Pedawa, Penglipuran, Legian) in Bali Province, Indonesia was conducted in 2008–2015 with written informed consent. Prior to the study, ethical approvals were obtained by the Eijkman Institute Research Ethics Commission (No. 32 on 27 October 2008 and No. 80 on 24 December 2014), and the Faculty of Medicine Ethic Committee, Udayana University (No. 690a/SKRT/X/2010 on 28 October 2010 and No. 1286/UN.14.2/Litbang/2014 on 18 September 2014).

The demographic data and anthropometric measurements were obtained including age, sex, body mass index (BMI) which was calculated by dividing weight (kg) by square of height ($m^2$), fasting plasma glucose (FPG), and serum lipid concentrations after overnight fasting for at least 10 h (triglyceride (TG), low-density lipoprotein-cholesterol (LDL-C), high-density lipoprotein cholesterol (HDL-C), total cholesterol (TC)], and triglyceride glucose (TyG) index which was calculated as Ln[fasting TG (mg/dL) $\times$ FPG (mg/dL)/2]. TyG index has been widely used as a surrogate marker for homeostasis of model assessment-insulin resistance (HOMA-IR) to evaluate insulin resistance, as both markers were positively correlated (*Aman et al., 2021*). Obesity was defined as having BMI $\geq$ 25 kg/m$^2$ (*World Health Organization, 2000*). Diabetes mellitus-FG was defined as having FPG $\geq$ 126 mg/dL (*American Diabetes Association, 2010*). Dyslipidemia was classified as having at least one of these following traits; high TG ($\geq$200 mg/dL), high LDL-C ($\geq$160 mg/dL), low HDL-C (<40 mg/dL), high TC ($\geq$240 mg/dL) (*NCEP, 2002*). Clinical dyslipidemia phenotypes following Fredrickson's classification were defined based on TG and TC levels, as follows: low TG, high TC (equal to IIa phenotype); high TG, low TC (equal to IV, I phenotype); high TG, high TC (equal to IIb, III, IV, V); and additional classification based on LDL-C and HDL-C levels, as follows: high LDL-C only; low HDL-C only (*Joint Committee for Guideline Revision, 2018*).

### DNA extraction and genotyping

Genomic DNA extraction was performed as described elsewhere (*Malik et al., 2011*). In this study, we selected two *TCF7L2* intronic SNPs, rs290487 and rs290481, which were common in East Asian (MAF > 0.60). Common SNPs were known to have lower false-positive rates than rare SNPs and produce more reliable results (*Tabangin, Woo & Martin, 2009*). The rs290487 and rs290481 were genotyped using the amplification refractory system mutation polymerase chain reaction (ARMS-PCR), a robust SNP genotyping method with high sensitivity (>80%) and specificity (>90%) (*Ye et al., 2001*; *Nanfack et al., 2015*), with novel set of primer pairs. Primer design was done using web-based primer design tool for ARMS-PCR, Primer1 (http://primer1.soton.ac.uk/primer1.html) (*Collins & Ke, 2012*) and edited using BioEdit® Sequence Alignment Editor (Ibis Bioscience, Carlsbad, CA, USA). Primer sequences are presented in Table S1.

The optimal annealing temperature was determined using the PCR gradient method (Bioline MyTaq™ HS DNA Polymerase) using the Veriti® thermal cycler (Applied Biosystem, Foster City, CA, USA), followed by visualization using 1% agarose gel electrophoresis (Lonza, Basel, Switzerland). Three samples that represented each genotype of rs290487 and rs290481 from ARMS-PCR results were randomly selected and confirmed by Sanger DNA sequencing using BigDye® Terminator v.3.1 Cycle Sequencing Kits, with ABI 3130xl Genetic Analyzer (Applied Biosystem, Foster City, CA, USA). Genotyping of rs290487 and rs290481 in DNA samples was performed using SimpliAmp™ Thermal Cycler (Applied Biosystem, Foster City, CA, USA) and resolved in 2% agarose gel electrophoresis (Lonza, Basel, Switzerland).

## Statistical analysis

Data analyses were carried out using R version 4.1.1 (www.r-project.org) with R Studio v1.4.1717 (www.rstudio.com). Continuous variables were presented as median (IQR) and compared by performing Wilcoxon-Mann Whitney U test. Categorical variables were presented as percentages and compared by performing Pearson's chi-squared test. The departure of genotype distribution from Hardy-Weinberg equilibrium was tested using Pearson's chi-squared test, and the $r^2$ and D' measure of linkage disequilibrium was evaluated, as implemented in the "genetics" package (*Warnes et al., 2021*). Age, sex, population, obesity, diabetes mellitus-FG, and high TyG index were used as adjustments for association analysis. The optimal cutoff point for a high TyG index (8.85) for dyslipidemia was analyzed using the Youden index in "OptimalCutpoints" package (*Lopez-Raton & Xose Rodriguez-Alvarez, 2021*). Adjusted odds ratios (ORs) with 95% confidence interval (95% CI) for associations of TCF7L2 SNPs with dyslipidemia and individual lipid profiles were estimated using the likelihood ratio test. Associations between dyslipidemia phenotypes based on Fredrickson's classification and each SNP were evaluated using multivariate multinomial logistic regression by implementing the "nnet" package, measured as adjusted relative risk ratios (RRRs) with 95% CI (*Ripley & Venables, 2022*). The association was significant when the $p$ value is < 0.025, following Bonferroni correction ($p$ value = 0.050/2 SNPs) (*Cheverud, 2001*). The adjusted ORs and 95% CI were illustrated as forest plot using "ggplot2" package (*Wickham et al., 2021*). Further, inferred haplotypes were estimated with the expectation maximization algorithm, as implemented in the "haplo.stats" package (*Sinnwell et al., 2021*). The haplotype associations with dyslipidemia and lipid profiles were determined using the generalized linear regression models, adjusted for age, sex, population, obesity, diabetes mellitus-FG, and TyG index. Empirical $p$ values ($p_{\text{sim}}$) at significant level 0.050 were calculated after 10,000 simulations for multiple testing correction (*Becker & Knapp, 2004*). Finally, the interaction analyses between *TCF7L2* SNPs and obesity status on dyslipidemia probability were investigated using the likelihood ratio test by considering all genetic models. The significant interactions ($p_{interaction}$ < 0.100) were then plotted using the "interactions" package (*Long, 2021*).

**Table 1 Baseline characteristics of non-dyslipidemic and dyslipidemic subjects.**

| Variable | Non-dyslipidemic ($n = 366$) | Dyslipidemic ($n = 199$) | $p$ |
|---|---|---|---|
| Age (years, median (IQR)) | 46.0 (40.0–56.0) | 47.0 (39.0–57.0) | 0.939 |
| Sex ($n$(%)) | | | |
| Female (244 (43.2)) | 191 (52.2) | 53 (26.6) | **<0.001** |
| Male (321 (56.8)) | 175 (47.8) | 146 (73.4) | |
| Age per Sex (years, median (IQR)) | | | |
| Female Age | 45.0 (40.0–55.5) | 52.0 (40.0–60.0) | 0.053 |
| Male Age | 46.0 (40.0–56.5) | 44.0 (38.0–54.8) | 0.317 |
| Population ($n$(%)) | | | |
| Rural | 200 (54.6) | 91 (45.7) | 0.053 |
| Urban | 166 (45.4) | 108 (54.3) | |
| BMI (kg/m$^2$, median (IQR)) | 22.9 (20.4–25.4) | 24.9 (22.4–28.2) | **<0.001** |
| FPG (mg/dL, median (IQR)) | 91.0 (83.3–99.0) | 93.0 (87.0–100.0) | **0.005** |
| TG (mg/dL, median (IQR)) | 97.5 (77.0–129.8) | 199.0 (138.5–237.0) | **<0.001** |
| LDL-C (mg/dL, median (IQR)) | 114.0 (95.3–133.0) | 135.0 (101.0–164.0) | **<0.001** |
| HDL-C (mg/dL, median (IQR)) | 54.0 (48.0–61.0) | 43.0 (37.5–52.0) | **<0.001** |
| TC (mg/dL, median (IQR) | 190.0 (170.0–201.8) | 225.0 (190.0–250.0) | **<0.001** |
| TyG index | 8.4 (8.1–8.7) | 9.1 (8.8–9.4) | **<0.001** |
| Disease prevalence ($n$(%)) | | | |
| Obesity | 107 (29.2) | 99 (49.7) | **<0.001** |
| Diabetes mellitus-FG | 17 (4.6) | 16 (8.0) | 0.145 |

Notes:
  IQR, interquartile range; BMI, body mass index; FPG, fasting blood glucose; TG, triglycerides; LDL-C, low-density lipoprotein cholesterol; HDL-C, high-density lipoprotein cholesterol; TC, total cholesterol; TyG, triglyceride and glucose. Criteria: obesity, BMI ≥ 25 kg/m$^2$ (*WHO, 2000*); diabetes mellitus-FG, FPG ≥ 126 mg/dL (*American Diabetes Association, 2010*); high TG, TG ≥ 200 mg/dL (*NCEP, 2002*); high LDL-C, LDL-C ≥ 160 mg/dL (*NCEP, 2002*); low HDL-C, HDL-C < 40 mg/dL (*NCEP, 2002*); high TC, TC ≥ 240 mg/dL (*NCEP, 2002*); dyslipidemia, the presence of at least one altered lipid profile (*NCEP, 2002*).
  The $p$ values were calculated using either Wilcoxon-Mann Whitney U test for continuous variables or Pearson's chi-squared test for categorical variables. The significant $p$ values are in bold ($p < 0.050$).

## RESULTS

### Baseline characteristics of the subjects

Characteristics of the study subjects are presented in Table 1. Subjects were grouped into dyslipidemic ($n = 366$) and non-dyslipidemic ($n = 199$) based on NCEP-ATP III criteria of high blood cholesterol (*NCEP, 2002*). Dyslipidemia was more prevalent in males than in females (73.4% *vs* 26.6%, $p < 0.001$). Compared to the non-dyslipidemic subjects, the dyslipidemic subjects had a significantly higher BMI, FPG, TG, LDL-C, TC levels, TyG index (all $p < 0.050$) and lower HDL-C levels ($p < 0.001$). The most prevalent type of dyslipidemia was high TG (49.7%), followed by high TC (41.2%), low HDL-C (37.2%), and high LDL-C (32.7%). Obesity was also prevalent among dyslipidemic subjects (49.7%) ($p < 0.001$).

### Genotypic and allelic distribution

The genotype and allele frequencies, Hardy-Weinberg equilibrium and linkage disequilibrium are shown in Table S2. The T alleles of rs290487 and rs290481, defined as a
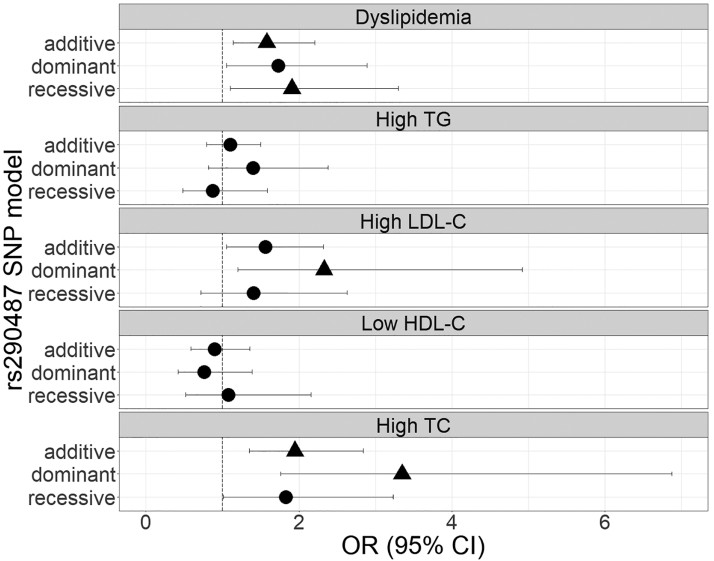

**Figure 1 Forest plot of odds ratios for the associations of rs290487 with dyslipidemia and altered lipid profiles.** Forest plot was generated using ggplot2 (▲: $p < 0.025$); OR (95% CI), Odds Ratio (95% Confidence Interval); TG, triglycerides; LDL-C, low-density lipoprotein cholesterol; HDL-C, high-density lipoprotein cholesterol; TC, total cholesterol. OR was adjusted by age, sex, population, obesity (BMI ≥ 25 kg/m²), diabetes mellitus-FG (FPG ≥ 126 mg/dL), and TyG index (except for high TG).       

minor allele according to dbSNP, were presented as the major alleles in Balinese, with the frequencies of 0.56 and 0.53, respectively. No significant departure from Hardy-Weinberg equilibrium was found for both SNPs ($p > 0.050$). High LD was found between rs290487 and rs290481 SNPs (D' = 0.90; $r^2$ = 0.72).

## Genetic associations with dyslipidemia and altered lipid profile

The genetic associations of rs290487 and rs290481 with dyslipidemia and lipid profile were presented in Figs. 1 and 2. Both SNPs were significantly associated with dyslipidemia and individual high TC levels. rs290487 was significantly associated with dyslipidemia in additive (OR 1.58, 95% CI [1.14–2.21], $p = 0.006$) and recessive model (OR 1.91, 95% CI [1.10–3.30], $p = 0.020$), meanwhile rs290481 was significantly associated in additive (OR 1.56, 95% CI [1.14–2.14], $p = 0.006$) and dominant model (OR 2.08, 95% CI [1.26–3.50], $p = 0.005$). Further, both SNPs were associated with high TC levels in additive (rs290487 OR 1.95, 95% CI [1.35–2.84], $p < 0.001$; rs290481 OR 1.78, 95% CI [1.22–2.56], $p = 0.002$) and dominant model (rs290487 OR 3.35, 95% CI [1.76–6.88], $p < 0.001$; rs290481 OR 2.79, 95% CI [1.48–5.69], $p = 0.003$). Additionally, rs290487 was also associated with high LDL-C levels in dominant model (OR 2.33, 95% CI [1.20–4.92], $p = 0.018$). Both SNPs did not show any significant associations with individual high TG and low HDL-C. Complete genotypic distribution of both SNPs on non-affected and dyslipidemia affected subjects are shown in Table S3. Further association analyses between each SNP and clinical dyslipidemia phenotypes according to Fredrickson's classification showed the similar effect on high TC, regardless of the TG level (Table 2). Both rs290487

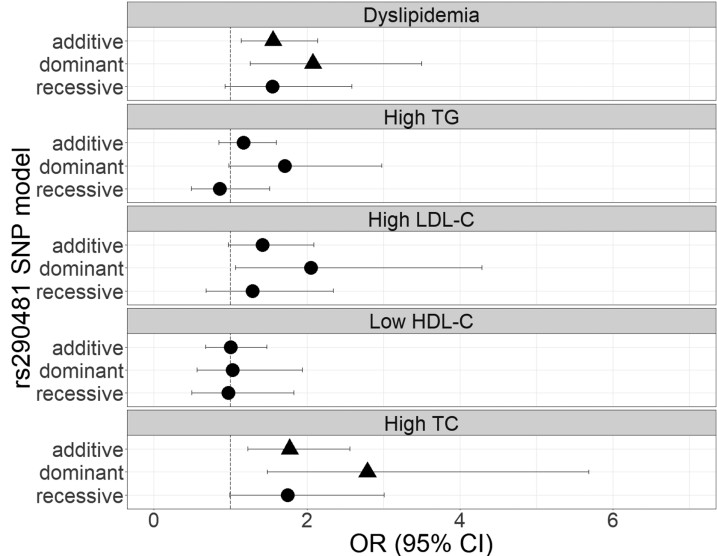

**Figure 2 Forest plot of odds ratios for the associations of rs290481 with dyslipidemia and altered lipid profiles.** Forest plot was generated using ggplot2 (▲: $p < 0.025$); OR (95% CI), Odds Ratio (95% Confidence Interval); TG, triglycerides; LDL-C, low-density lipoprotein cholesterol; HDL-C, high-density lipoprotein cholesterol; TC, total cholesterol. OR was adjusted by age, sex, population, obesity (BMI ≥ 25 kg/m$^2$), diabetes mellitus-FG (FPG ≥ 126 mg/dL), and TyG index (except for High TG).

and rs290481 were significantly associated with increase odds for developing combined low TG, high TC (equal to IIa phenotype) (rs290487 additive OR 1.65, 95% CI [1.07–2.55], $p = 0.023$; rs290481 additive OR 1.67, 95% CI [1.09–2.56], $p = 0.018$) and high TG, high TC (equal to IIb, III, IV, V phenotypes) (rs290487 additive OR 2.43, 95% CI [1.33–4.43], $p = 0.004$, dominant OR 17.02, 95% CI [2.23–130.89], $p = 0.007$; rs290481 additive OR 2.01, 95% CI [1.12–3.61], $p = 0.019$, dominant OR 6.99, 95% CI [1.59–30.83], $p = 0.010$).

## *TCF7L2* rs290487 and rs290481 haplotype association with dyslipidemia and altered lipid profile

Since the two SNPs had a high LD, we therefore conducted haplotype association analysis with dyslipidemia and altered lipid profile, as shown in Table 3. Only haplotypes with a frequency of ≥0.1 were analyzed. We identified one haplotype, CC, carrying risk alleles from both SNPs, was strongly associated with dyslipidemia, high LDL-C, and high TC ($p_{sim} < 0.050$) under dominant and additive models.

## The interactions between *TCF7L2* rs290487 and rs290481 with obesity towards dyslipidemia probability

To investigate the modulatory effect of obesity in influencing SNPs associations with dyslipidemia, we analyzed the interaction between rs290487 and rs290481 SNPs and obesity status by considering all genetic models (Table S4). As shown in Fig. 3, obesity modified the effect of rs290487 and rs290481 genotypes on dyslipidemia risk. In the additive model for both SNPs, subjects carrying the heterozygous CT genotype were more

**Table 2 Multinomial logistic regression results for associations of rs290487 and rs290481 with dyslipidemia phenotypes.**

| Dyslipidemia phenotype | Equal to Fredrickson's classification | N | Genotype frequency | | | Additive | | Dominant | | Recessive | |
|---|---|---|---|---|---|---|---|---|---|---|---|
| | | | TT | TC | CC | RRR [95% CI] | $p$ | RRR [95% CI] | $p$ | RRR [95% CI] | $p$ |
| **rs290487** | | | | | | | | | | | |
| Non-dyslipidemic | | 366 | 0.21 | 0.47 | 0.33 | Reference | | Reference | | Reference | |
| Low TG, High TC | IIa | 53 | 0.30 | 0.51 | 0.19 | 1.65 [1.07–2.55] | **0.023** | 1.96 [0.95–4.03] | 0.069 | 1.90 [0.96–3.77] | 0.066 |
| High TG, Low TC | IV, I | 70 | 0.19 | 0.56 | 0.26 | 0.92 [0.61–1.38] | 0.673 | 0.93 [0.52–1.67] | 0.805 | 0.84 [0.39–1.79] | 0.647 |
| High TG, High TC | IIb, III, IV, V | 29 | 0.28 | 0.66 | 0.07 | 2.43 [1.33–4.43] | **0.004** | 17.02 [2.23–130.89] | **0.007** | 1.70 [0.66–4.35] | 0.271 |
| High LDL-C only | | 8 | 0.25 | 0.63 | 0.13 | 1.86 [0.63–5.44] | 0.260 | 3.41 [0.39–29.63] | 0.266 | 1.66 [0.32–8.71] | 0.552 |
| Low HDL-C only | | 39 | 0.28 | 0.44 | 0.28 | 1.19 [0.71–2.02] | 0.510 | 0.86 [0.41–1.82] | 0.692 | 1.93 [0.86–4.35] | 0.110 |
| **rs290481** | | | | | | | | | | | |
| Non-dyslipidemic | | 366 | 0.21 | 0.47 | 0.33 | Reference | | Reference | | Reference | |
| Low TG, High TC | IIa | 53 | 0.30 | 0.51 | 0.19 | 1.67 [1.09–2.56] | **0.018** | 2.07 [0.99–4.33] | 0.052 | 1.84 [0.96–3.55] | 0.067 |
| High TG, Low TC | IV, I | 70 | 0.19 | 0.56 | 0.26 | 1.08 [0.73–1.58] | 0.702 | 1.36 [0.74–2.51] | 0.321 | 0.84 [0.43–1.68] | 0.631 |
| High TG, High TC | IIb, III, IV, V | 29 | 0.28 | 0.66 | 0.07 | 2.01 [1.12–3.61] | **0.019** | 6.99 [1.59–30.83] | **0.010** | 1.46 [0.59–3.62] | 0.419 |
| High LDL-C only | | 8 | 0.25 | 0.62 | 0.13 | 1.54 [0.55–4.32] | 0.409 | 3.27 [0.39–27.68] | 0.278 | 1.12 [0.21–5.95] | 0.898 |
| Low HDL-C only | | 39 | 0.28 | 0.44 | 0.28 | 1.26 [0.77–2.07] | 0.359 | 1.14 [0.53–2.44] | 0.739 | 1.58 [0.73–3.42] | 0.249 |

Notes:

RRR, relative risk ratio; 95% CI, 95% confidence interval; TG, triglycerides; LDL-C, low-density lipoprotein cholesterol; HDL-C, high-density lipoprotein cholesterol; TC, total cholesterol. Altered lipid profiles criteria: high TG (≥200 mg/dL), high LDL-C (≥160 mg/dL), low HDL-C (<40 mg/dL), high TC (≥240 mg/dL) (*NCEP, 2002*).
Dyslipidemia phenotypes were classified as follows: type IIa: high TC and low TG; type IV and I: high TG and low TC; type IIb, III, IV, and V: high TG and high TC, following the Fredrickson's classification (*Joint Committee for Guideline Revision, 2018*); with additional high LDL-C only and low HDL-C only groups.
Analyses were performed using multivariate multinomial logistic regression, adjusting for age, sex, population, obesity (BMI ≥ 25 kg/m$^2$ (*WHO, 2000*)) and diabetes mellitus-FG (FPG ≥ 126 mg/dL (*American Diabetes Association, 2010*)).
The significant $p$ values after Bonferroni's correction ($p < 0.025$) are in bold.

**Table 3 Association of *TCF7L2* haplotype with dyslipidemia and altered lipid profile.**

| Trait | Haplotype | | Frequency | | Additive | | | | Dominant | | | |
|---|---|---|---|---|---|---|---|---|---|---|---|---|
| | rs290487 | rs290481 | NAS | AS | SS | $p_{sim}$ | OR [95% CI] | $p$ | SS | $p_{sim}$ | OR [95% CI] | $p$ |
| Dyslipidemia | T | T | 0.53 | 0.46 | −2.674 | **0.007** | Reference | | −1.980 | 0.051 | Reference | |
| | C | C | 0.39 | 0.47 | 2.918 | **0.002** | 1.62 [1.16–2.26] | **0.003** | 2.650 | **0.007** | 1.93 [1.17–3.19] | **0.008** |
| High TG | T | T | 0.52 | 0.48 | −0.676 | 0.488 | Reference | | 0.607 | 0.560 | Reference | |
| | C | C | 0.41 | 0.45 | 0.812 | 0.421 | 1.14 [0.81–1.61] | 0.416 | 1.470 | 0.144 | 1.47 [0.87–2.49] | 0.141 |
| High LDL-C | T | T | 0.52 | 0.43 | −1.908 | 0.067 | Reference | | −0.928 | 0.335 | Reference | |
| | C | C | 0.41 | 0.51 | 2.125 | **0.029** | 1.52 [1.02–2.28] | **0.033** | 2.363 | **0.015** | 2.19 [1.11–4.29] | **0.018** |
| Low HDL-C | T | T | 0.51 | 0.50 | −0.004 | 1.000 | Reference | | −0.008 | 0.991 | Reference | |
| | C | C | 0.42 | 0.40 | −0.474 | 0.665 | 0.94 [0.62–1.44] | 0.635 | −0.777 | 0.450 | 0.83 [0.46–1.51] | 0.437 |
| High TC | T | T | 0.53 | 0.40 | −3.376 | **<0.001** | Reference | | −2.350 | **0.020** | Reference | |
| | C | C | 0.40 | 0.53 | 3.413 | **<0.001** | 1.94 [1.33–2.85] | **<0.001** | 3.650 | **<0.001** | 3.50 [1.79–6.84] | **<0.001** |

Notes:

NAS, Non-Affected Subjects; AS, Affected Subjects; SS, Score Statistics; OR, odds ratio; 95% CI, 95% confidence interval; TG, triglycerides; LDL-C, low-density lipoprotein cholesterol; HDL-C, high-density lipoprotein cholesterol; TC, total cholesterol.
Dyslipidemia was defined by the presence of one of the following criteria: high TG (≥200 mg/dL), high LDL-C (≥160 mg/dL), low HDL-C (<40 mg/dL) or high TC (≥240 mg/dL) (*NCEP, 2002*). Haplotypes with frequency of ≥0.1 were included in the analysis. Association analysis was performed using the adjusted likelihood ratio test, by assuming additive and dominant genetic model and controlling for age, sex, population, obesity (BMI ≥ 25 kg/m$^2$ (*WHO, 2000*)), diabetes mellitus-FG (FPG ≥ 126 mg/dL (*American Diabetes Association, 2010*)), and TyG index (except for High TG).
The $p_{sim}$ is a simulated $p$ value after minimal 10,000 simulations.
The significant $p$ values ($p < 0.050$) are in bold.

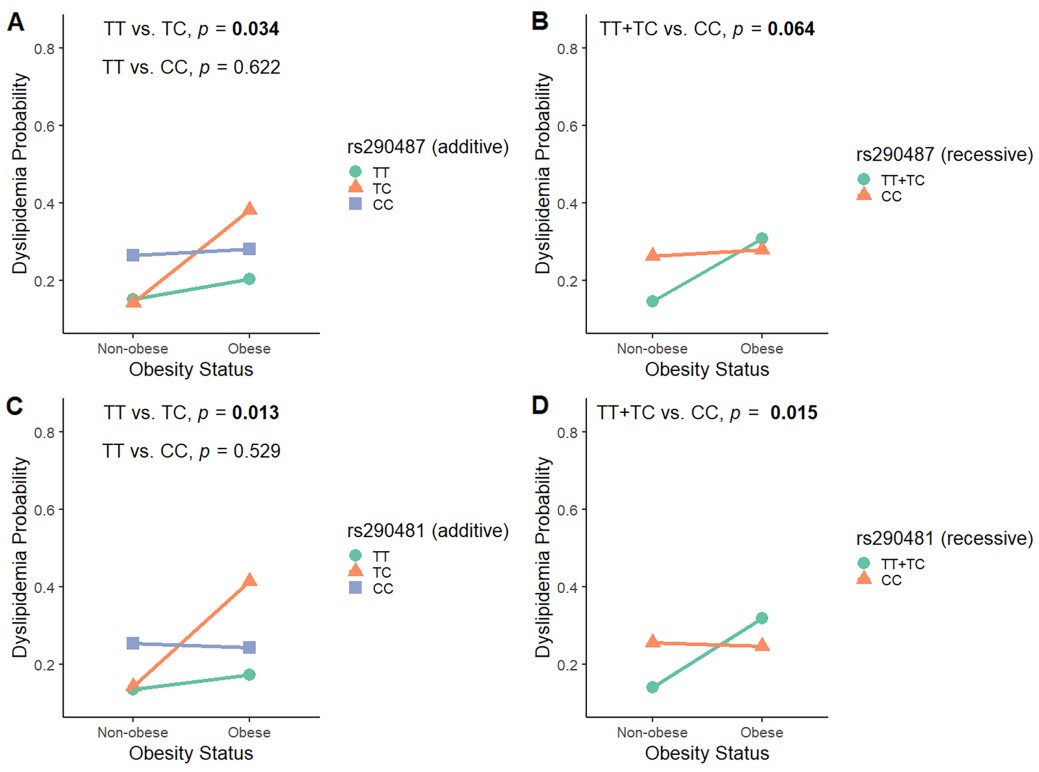

**Figure 3 Interaction plot between *TCF7L2* SNPs and obesity status on dyslipidemia risk.** Interaction plot between rs290487 and obesity status on dyslipidemia risk under additive (A) and recessive (B) model. Interaction plot between rs290481 and obesity status on dyslipidemia risk under additive (C) and recessive (D) model.

likely to develop dyslipidemia when they had obesity, when compared to the other genotypes carriers (rs290487 $p_{interaction}$ = 0.034; rs290481 $p_{interaction}$ = 0.013). When being analysed under the recessive model, the TT+TC genotypes carrier exhibited an increased dyslipidemia probability when co-exist with obesity (rs290487 $p_{interaction}$ = 0.064; rs290481 $p_{interaction}$ = 0.015).

## DISCUSSION

Our study aimed to investigate the association of *TCF7L2* intronic SNPs rs290487 and rs290481 with dyslipidemia in Balinese population. The SNPs' influence on metabolic abnormalities in the selected Indonesian population has not been well-described. In this study, we reported the notable impacts of individual genotypes and a haplotype of the rs290487 and rs290481 SNPs on dyslipidemia and individual lipid profile in the Balinese. Over the past three decades, this population has been exposed to westernization and urbanization brought by rapid tourism development, leading to a high prevalence of obesity and metabolic syndrome, particularly in tourism destination area (*Suastika et al., 2011*). Our findings support the notion that *TCF7L2* SNPs have a significant impact on lipid metabolism.

The rs290487 and rs290481 allele frequencies varied among different populations. In East Asians, these SNPs were commonly found on average MAF of 0.60 (*Chang et al., 2007*;
*Liu et al., 2009*; *Wang et al., 2013*; *Zhu et al., 2017*), but found in much lower frequencies among Caucasians (*Delgado-Lista et al., 2010*), the population referred by dbSNP. To the best of our knowledge, this study is the first report on MAF of rs290487 and rs290481 in the Balinese population. Our findings showed the frequencies of T allele of rs290487 and rs290481 were 0.56 and 0.53, respectively, while C allele of both SNPs automatically became the minor allele. Differences in allelic distribution can occur due to some conditions, such as different genetic background between populations (*Ding & Kullo, 2011*) and natural selection mechanism, whereas frequency of adaptive alleles in a population tend to be arisen (*Kido et al., 2018*).

In this study, we found dyslipidemia more prevalent in males than females. Changes towards unhealthy lifestyles in the Balinese increase the risk of dyslipidemia in this population. According to the Bali Province Basic Health Survey 2018, Balinese men relatively consumed more sweetened beverages (including energy drink and soft drink) and less vegetables and fruits than women (*Kementerian Kesehatan, 2019b*). In general, risks of dyslipidemia in men were dominantly associated with unhealthy lifestyles, such as high fatty and salty food intake, smoking, hypertension, obesity, and diabetes (*Wang et al., 2020*; *Xi et al., 2020*). Meanwhile, dyslipidemia in women was often found at postmenopausal state due to decreased in estrogen level and its lipoprotein maintenance role (*Reddy Kilim & Rao Chandala, 2013*; *Opoku et al., 2019*; *Xi et al., 2020*). Unfortunately, lifestyle and physiological data from dyslipidemic subjects were not available, which is the limitation of this study.

Our findings showed that the *TCF7L2* rs290487 and rs290481 C alleles were significantly associated with dyslipidemia and high TC levels. Additional association was found between rs290487 and high LDL-C levels. These associations were consistently found in haplotype CC from both SNPs. Furthermore, we did not find any significant associations between rs290487 and rs290481 with individual TG levels, even though our dyslipidemic subjects exhibited 2-fold higher TG levels, compared to non-dyslipidemic subjects. Interestingly, the associations of both SNPs with either low or high TG levels were only detected when it appeared together with high TC levels, indicating a stronger association of these SNPs with TC rather than TG level itself. TC level reflects the total amount of cholesterol, including LDL-C, HDL-C, and other lipids. Therefore, the significant association between *TCF7L2* SNPs with TC might occur due to its major influence on LDL-C. A study performed using MetS subject from eight European countries have reported that rs290481 C allele, the major allele in that population, also contributed to higher LDL-C levels (*Delgado-Lista et al., 2010*). Our previous study had also reported the associations of three *TCF7L2* SNPs (rs7903146, rs12255372, rs10885406) with elevated TC/HDL-C ratio (*Oktavianthi et al., 2018*). TC levels was known to be increased primarily due to elevated LDL-C levels (*Kreisberg & Kasim, 1987*). High LDL-C and TC level might increase cardiovascular disease risk by promoting atherosclerosis process (*Hedayatnia et al., 2020*).

As a wide-range transcription factor, TCF7L2 takes part in regulating the gene expression by binding to the promoter of its target genes (*Norton et al., 2014*; *Zhao et al., 2016*). The presence of intronic polymorphisms may interrupt the alternative splicing

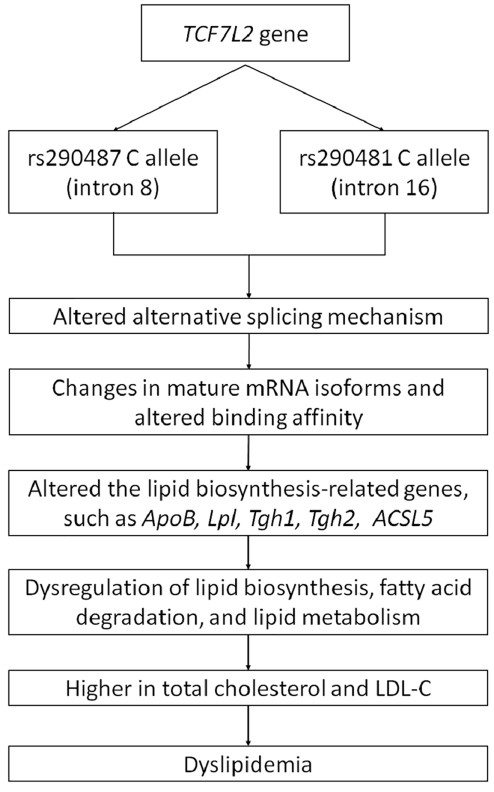

**Figure 4 Conceptual framework depicting relationship between *TCF7L2* SNPs with dyslipidemia.**

mechanism of *TCF7L2*, resulting in changes of mRNA isoforms and leading to dysregulation of its downstream target genes (*Mondal et al., 2010*; *Buroker, 2017*). *Pradas-Juni et al. (2014)* have found an increase of exon four-containing mRNA transcripts in pancreatic islets of T2DM patient with rs7903146 TT genotype. The inclusion or exclusion of exon 4 might be influenced by the presence of rs7903146 in intron 4. On the other hand, rs290487 and rs290481 which are located in intron 8 and 16, respectively, might possess different approaches in alternative splicing regulation (*Liu et al., 2009*). The transition from T to C in rs290487 was suggested to induce altered binding affinity of TCF7L2 binding sites to its target genes (*Zhang et al., 2020*). Many studies have demonstrated the role of TCF7L2 in regulating genes that are involved in cholesterol and triglyceride biosynthesis, such as *ApoB* (*Norton et al., 2014*), *Lpl* (*He et al., 2020*), *Tgh1*, and *Tgh2* (*Geoghegan et al., 2019*). *Geoghegan et al. (2019)* have also found TCF7L2 binding sites within 1 kb of the promoter of numerous genes involved in the *de novo* lipogenesis pathway. Moreover, a specific *TCF7L2* rs7903146 region was known to control *ACSL5* gene expression which is important for lipid biosynthesis and fatty acid degradation by interacting with its promoter as an enhancer (*Xia et al., 2016*). Therefore, the effect of intronic variants within the *TCF7L2* gene on dyslipidemia might be done through several pathways. A conceptual framework for summarizing the relationship between *TCF7L2* SNPs and dyslipidemia was proposed (Fig. 4).

In the current study, we found that obesity modulates the SNPs' association with dyslipidemia, particularly in those who carry heterozygous genotype. Previously, the modulatory effect of obesity in genetic risks towards dyslipidemia has also been reported (*Yin et al., 2012*; *Cole, Nikpay & McPherson, 2015*). Obesity and dyslipidemia are linked by insulin resistance, as reviewed in *Vekic et al. (2019)*. Since we lacked HOMA-IR data, the most common insulin resistance marker, we used TyG index as a surrogate marker (*Simental-Mendía, Rodríguez-Morán & Guerrero-Romero, 2008*; *Khan et al., 2018*; *Aman et al., 2021*). Although the TyG index is a significant determining factor for dyslipidemia, we did not find any direct impact of rs290487 and rs290481 on it (Table S5). This might suggest that the SNPs are rather influencing lipid metabolism. Further studies are warranted to explore rs290487 and rs290481 influence on the other lipid fractions (VLDL, chylomicron, ApoE, etc.), to understand on how *TCF7L2* gene polymorphisms influence on dyslipidemia.

## CONCLUSIONS

In this study, we have shown that *TCF7L2* rs290487 and rs290481 C allele were significantly associated with dyslipidemia, high LDL-C, and high TC in Balinese population. Despite being minor alleles, the C alleles of both SNPs were relatively high, which raises the assumption that more than 40% of the Balinese were very likely to carry this allele. In addition, the changes towards unhealthy lifestyles (high calorie intake and sedentary lifestyle) increased the Balinese's risk of dyslipidemia even more. As SNPs were unmodifiable risk, lifestyle improvement is needed to lower the risk of dyslipidemia, especially in the Balinese. The Indonesian population is very diverse, therefore, future studies to explore these associations in different populations in Indonesia should be conducted.

## ACKNOWLEDGEMENTS

This study was conducted as a collaborative initiative between Department of Internal Medicine, Udayana University/Sanglah Hospital, Denpasar, Bali and Eijkman Institute for Molecular Biology, Jakarta. The authors sincerely acknowledge the participation and support of all volunteers, field medical doctors, medical faculty students, clinical pathology laboratory and research assistants. We thank Pande Dwipayana, Desak Made Wihandani, I Wayan Weta for their support during sample collections and Dr. Ni Luh Made Agustini Leonita and Clarissa Asha Febinia for their help in DNA isolation.

### Funding

This research was supported by a block grant from the Government of Republic of Indonesia through the Ministry of Research and Technology for the Eijkman Institute for Molecular Biology. The funders had no role in study design, data collection and analysis, decision to publish, or preparation of the manuscript.

## Grant Disclosures

The following grant information was disclosed by the authors:

Government of Republic of Indonesia through the Ministry of Research and Technology for the Eijkman Institute for Molecular Biology.

## Competing Interests

The authors declare that they have no competing interests.

## Author Contributions

- Prisca C Limardi performed the experiments, analyzed the data, prepared figures and/or tables, authored or reviewed drafts of the paper, and approved the final draft.
- Sukma Oktavianthi conceived and designed the experiments, analyzed the data, prepared figures and/or tables, authored or reviewed drafts of the paper, and approved the final draft.
- Lidwina Priliani conceived and designed the experiments, analyzed the data, prepared figures and/or tables, authored or reviewed drafts of the paper, and approved the final draft.
- Retno Lestari analyzed the data, authored or reviewed drafts of the paper, and approved the final draft.
- Made Ratna Saraswati analyzed the data, authored or reviewed drafts of the paper, and approved the final draft.
- Ketut Suastika analyzed the data, authored or reviewed drafts of the paper, and approved the final draft.
- Safarina G Malik conceived and designed the experiments, analyzed the data, authored or reviewed drafts of the paper, and approved the final draft.

## Human Ethics

The following information was supplied relating to ethical approvals (*i.e.*, approving body and any reference numbers):

The Eijkman Institute Research Ethics Commission (No. 32 on 27 October 2008 and No. 80 on 24 December 2014)

and

The Faculty of Medicine Ethic Committee, Universitas Udayana (No. 690a/SKRT/X/ 2010 on 28 October 2010 and No. 1286/UN.14.2/Litbang/2014 on 18 September 2014).

## Field Study Permissions

The following information was supplied relating to field study approvals (*i.e.*, approving body and any reference numbers):

The Faculty of Medicine Ethic Committee, Udayana University for field work.

The following information was supplied relating to field study approvals (*i.e.*, approving body and any reference numbers):

We received permission from the following leaders of villages and regencies: I Made Madia Suryanatha S.S.T.P (Legian Village of the Badung Regency), I Wayan Supat

(Penglipuran Village of the Bangli Regency), Ketut Gede Arjaya (Nusa Ceningan Village of the Klungkung Regency) and I Putu Sudarmaja (Pedawa Village of the Buleleng Regency).

## Data Availability

The data is available at Mendeley: Priliani, Lidwina (2021), "TCF7L2 dataset", Mendeley Data, V1, DOI 10.17632/xv5jwz3hhv.1.

## Supplemental Information

Supplemental information for this article can be found online at http://dx.doi.org/10.7717/peerj.13149#supplemental-information.

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
