# Peer review of "Transcription factor 7-like 2 single nucleotide polymorphisms rs290487 and rs290481 are associated with dyslipidemia in the Balinese population"

_PeerJ, doi:10.7717/peerj.13149_

## Round 0.1 · original submission · Major Revisions

The authors should address all the concerns raised by the reviewers.

Reviewer 1 ·

Basic reporting

The article was written by Safarina G Malik, et al is a valuable article and provides useful information, However, there are some issues that can help improve the work and can be accepted after solving these issues.
This study aimed to investigate the association of TCF7L2 intronic SNPs rs290487 and rs290481 with dyslipidemia in the Balinese population. The SNPs’ influence on metabolic abnormalities in selected Indonesian population. The findings support the notion that TCF7L2 SNPs have a significant impact on lipid metabolism.
a. Should mention about the genetic model in result/discussion part whether how related to dyslipidemia
b. Subgroup Analysis of dyslipidemia with or without diabetes and obesity as TCF7L2 SNPs associated with metabolic diseases.

Experimental design

The manuscript is well written and technically sound. However, there are some points to be considered in the revision of the manuscript:
a. The rationale of SNP selection would be elaborated.
b. Define the metabolic disease (diabetes/obesity) in section subject and as in table 1.
c. % Error of ARMS-PCR method and mention how random genotyping confirmed by Sanger DNA sequencing.

Validity of the findings

In this study, the author reported the impacts of individual genotypes and a haplotype of the rs290487 and rs290481 SNPs on dyslipidemia and individual lipid profile in the Balinese. This population has been exposed to westernization and urbanization brought by rapid tourism development.

It would be more interesting if the authors mention the genotype/haplotype comparing other populations especially in the Asian population and related to tourism as mentioned in the conclusion part.

Additional comments

To make this manuscript much readable, some key data from the experiment could be considered to reorganize as graphics in Figure 3, such as rs290481 genotype and lipid profile level and some proposed mechanisms.

Reviewer 2 ·

Basic reporting

This clinical study has investigated the associations two SNPs of transcription factor 7-like 2 (TCF7L2) gene viz. rs290487 and rs290481 with altered lipid profile in the Balinese population. The authors have done SNP genotyping and serum lipid profile analysis from 565 subjects from four locations in Bali province. They report existence of significant associations of these TCF7L2 polymorphisms with dyslipidemia and altered lipid profile in these individuals. Although the study appears simplistic and lacks translational value, the clinical findings reported appear to be novel and have the merit for publication. There are several concerns that need to be addressed before the study can be considered for publication in its current form.

Experimental design

One of the major shortcomings of the publication is that the SNPs are associated with dyslipidemia in general which can be broad and cover a number of diseases. The authors are recommended to use clinically used Frederickson classification of dyslipidemias and associate each of these SNPs to the specific type of dyslipidemia. This will greatly add to the value of this publication and shed more light on the specificity of the genetics involved.

Validity of the findings

No comments

Additional comments

Major comments:
• The authors have reported no significant association of SNP rs290487 with dyslipidemia. Hence, the title appears to be misleading for the future readers and should be revised accordingly to include either only rs290481 or change to specific significant associations.
• Table 1: n(%) for Sex must be reported separately as %of total Males and %of total females separately for two reasons: 1) Total number of Males used in the study is much higher 2) Percentage of dyslipidemic subjects compared to percentage of non-dyslipidemic subjects appears to be lower in both males and females. This will add clarity for future readers.
• Looking at the baseline characteristics, it appears that TG levels are the most altered (over 2-fold increase) in dyslipidemic individuals studied compared to levels of other lipids. This underscores the importance of studying the mechanistic role of these TCF7L2 SNPs in triglyceride metabolism and/or storage. Are there any prior studies where these SNPs or other mutations in TCF7L2 gene have been shown to play a role in the pathogenesis of hypertriglyceridemia? If yes, the authors are encouraged to cite them. If not, the authors are encouraged to expand on this interesting observation and comment in the discussion section.
• The authors do mention a limitation of being unable to measure HOMA-IR index. However, if the authors have access to blood samples they are encouraged to measure and report HOMA-IR index in the revised submission. It would address a very critical question of the involvement of the degree of insulin resistance in these individuals.
Minor comments:
• Table 1: Please report Age for Males and Females separately
• Line 224: Please change ‘specific adipocyte’ to ‘adipocyte-specific’
• The paper can use a thorough revision of grammatical errors throughout the discussion text
• Table 2: The authors are recommended to present significant associations in the forms of Manhattan plots, this will give a better idea of the variability in the data

---

## Round 0.2 · accepted · Accept

The authors have extensively revised their manuscript, and satisfactorily addressed all the concerns raised by the reviewers.

Reviewer 2 ·

Basic reporting

No further comments

Experimental design

No further comments

Validity of the findings

No further comments